# MFDGCN: Multi-Stage Spatio-Temporal Fusion Diffusion Graph Convolutional Network for Traffic Prediction

Zhengyan Cui [1], Junjun Zhang [1], Giseop Noh [2] and Hyun Jun Park [2,*]

1    Department of Computer Information Engineering, Cheongju University, Cheongju 28503, Korea;
     cuizy1017@gmail.com (Z.C.); zjj416320@gmail.com (J.Z.)
2    Division of Software Convergence, Cheongju University, Cheongju 28503, Korea; kafa46@cju.ac.kr
*    Correspondence: hyunjun@cju.ac.kr

**Abstract:** Traffic prediction is a popular research topic in the field of Intelligent Transportation System (ITS), as it can allocate resources more reasonably, relieve traffic congestion, and improve road traffic efficiency. Graph neural networks are widely used in traffic prediction because they are good at dealing with complex nonlinear structures. Existing traffic prediction studies use distance-based graphs to represent spatial relationships, which ignores the deep connections between non-adjacent spatio-temporal information. The use of a simple approach to fuse spatio-temporal information is not conducive to obtaining long-term deep spatio-temporal dependencies. Therefore, we propose a new deep learning model Multi-Stage Spatio-Temporal Fusion Diffusion Graph Convolutional Network (MFDGCN). It generates multiple static and dynamic spatio-temporal association graphs to enhance features and adopts the multi-stage hybrid spatio-temporal fusion method. This promotes the effective fusion of a spatio-temporal multimodal and uses the diffuse convolution method to model the graph structure and time series in traffic prediction, respectively. The model can better predict both long and short-term traffic simultaneously. We evaluated MFDGCN using real road network traffic data and it shows good performance.

**Keywords:** traffic prediction; spatio-temporal prediction; graph convolutional network; temporal convolutional network; multi-head attention

## 1. Introduction

Traffic prediction is an important part of the Intelligent Transportation System (ITS), which can reasonably allocate road resources, alleviate traffic congestion, and improve road traffic efficiency [1,2]. Traffic prediction has always been extremely challenging because traffic in road networks changes dynamically over time. There is a complex, nonlinear and multimodal spatio-temporal dependency between historical traffic and predicted traffic. This temporal dependence relationship is expressed as the mutual influence of traffic at different times within the road network. The spatial dependence relationship is expressed as the mutual influence of the traffic between different roads within the road network, as shown in Figure 1.

With the vigorous development of neural networks, deep learning has achieved success in obtaining complex nonlinear relationships [3]. Graph Convolutional Networks (GCNs) are good at processing nonlinear structural data and are widely used in traffic prediction [4–7]. Most existing traffic prediction studies are aimed at short-term prediction (15 min) [1,8–10]. DCRNN [8] uses the graph convolution operation to replace the original linear transformation in the recursive unit of the Recurrent Neural Network (RNN) for obtaining spatio-temporal information. To avoid the complex gating operation of RNN, STGCN [9] uses graph convolution and 1D convolution to obtain spatio-temporal information, reduce parameters, and achieve a good short-term prediction effect. GWnet [10] further uses graph convolution and block stacking of the Temporal Convolution Network

(TCN) to achieve better short-term prediction. STSGCN [11] constructs a local spatio-temporal graph for convolution to obtain a better short-term prediction effect. GMAN [12] pointed out that the existing models pay more attention to short-term prediction and therefore spatio-temporal attention and an encoder–decoder mechanism were used to obtain better long-term predictions (1 h). However, this did not significantly help with short-term predictions. To be able to consider the combined effect of both short-term and long-term traffic prediction, we propose a new deep learning model Multi-Stage Spatio-Temporal Fusion Diffusion Graph Convolutional Network (MFDGCN).

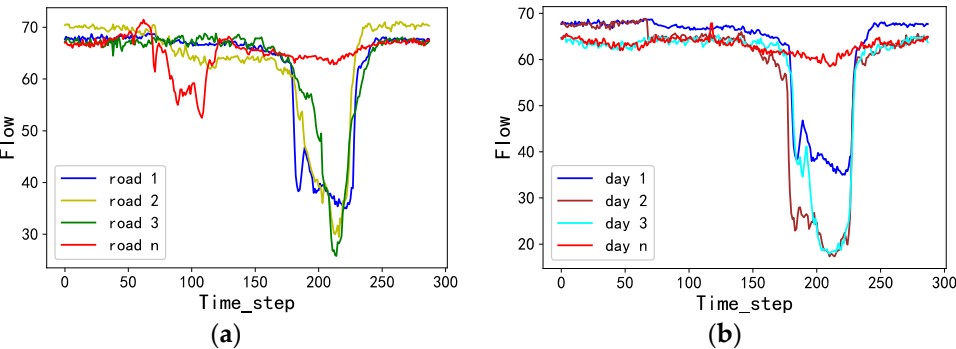

**Figure 1.** Traffic flow on different roads and times in a day. (**a**) At a specific time, roads 1, 2, 3 represent the traffic flow of three adjacent sensors and road *n* represents the traffic flow of a non-adjacent sensor; (**b**) at a specific sensor, days 1, 2, 3 represent the traffic flow of three adjacent days and day *n* represents the traffic flow of a non-adjacent day.

First, we propose a new spatio-temporal multi-association graph generation method, which obtains the static and dynamic representations of traffic flow in the spatial dimension based on spatial distance and spatial similarity. We then obtain the static and dynamic representation of the temporal dimension per the temporal connection relationship and temporal similarity. However, modeling based on a simple distance graph [8–10] will ignore the deep relationship between non-adjacent spatial and temporal dimensions. It will also be less able to capture the interaction of traffic between different roads. While using weather and POI (Point of Interest) as features to build models is possible [1,13], these data are not easy to obtain.

Second, we propose a multi-stage hybrid spatio-temporal fusion method that can capture longer-term spatio-temporal relationships. It uses basic fusion operations to capture early low-level spatio-temporal information, followed by the use of an adaptive gating mechanism to fuse high-level spatio-temporal information after convolution. It then finally captures deeper complex spatio-temporal information through a multi-head attention mechanism [14]. The spatial and temporal data in traffic prediction can be regarded as multimodal data with different dimensions. However, simple connection or addition operations are not sufficient to deeply mine for information from the mutual fusion of different models [8–10,15]. For example, STGCN [9] and HetGAT [16] use separate modules to sequentially capture spatial and temporal relationships, thereby splitting the association between both dimensions. STGODE [17] and AST-MTL [18] add the separately obtained spatio-temporal relationships, which is not conducive to establishing long-term spatio-temporal fusion. The main contributions of this paper are summarized as follows:

1. We propose a new spatio-temporal multi-association graph generation method to enhance features and capture richer spatio-temporal static and dynamic traffic features;
2. Considering the complex and changing spatio-temporal information of the traffic network, we propose a multi-stage hybrid spatio-temporal fusion method, which can capture longer-term spatio-temporal relationships;
3. We propose a new deep learning model MFDGCN, which can not only predict short-term traffic well, but also achieves good performance in long-term prediction;

4.   We perform extensive validation using data from a real road network, and our model MFDGCN shows better performance in long and short-term traffic prediction compared to existing advanced baselines.

## 2. Related Work

Spatio-temporal prediction refers to the prediction of unknown system states in spatial and temporal dimensions and is widely used in many real-world scenarios, such as weather prediction, traffic prediction, and earthquake prediction. Traffic prediction is a typical spatio-temporal prediction problem [2,9]. Traditional traffic prediction utilized model-driven methods, such as Historical Average (HA) [19] and Autoregressive Integrated Moving Average (ARIMA). These models are based on the linear analysis method, which are less accurate in complex nonlinear traffic prediction. Subsequently, data-driven methods gradually became mainstream, and machine learning methods were used in the early days, such as Vector Auto Regression (VAR) [20], Support Vector Regression (SVR) [21], and K-Nearest Neighbor (KNN) [22], but they required more detailed engineering features and were more complex in terms of time. With the rapid development of deep learning technology, Recurrent Neural Networks (RNNs) have been widely used in time series prediction tasks [23–25]. However, RNNs cannot consider the correlation of nodes in the spatial dimension. As such, Convolution Neural Networks (CNNs) for image processing have become a popular research topic, which divides the space into grids for convolution to capture spatial traffic dependencies [26]. However, regularized grid data is unable to properly reflect irregular traffic network structure information. Therefore, Graph Convolutional Networks (GCNs) [4–7] have received extensive attention and are widely used in traffic prediction to extract spatial dependencies.

GCN is divided into spectral domain graph convolution [4] and spatial domain graph convolution [5–7,27]. The spectral domain graph convolution uses Fourier transform to convert the graph signal to the spectral domain, followed by the convolution operation with a rather complicated calculation. The spatial domain graph convolution directly defines the neighborhood nodes of the convolution on the graph followed by the convolution operation, which is more intuitive and flexible. For example, GraphSage [6] introduces an aggregation function to aggregate the neighbor information of nodes. GAT [7] uses an attention mechanism to determine the importance of each neighbor node. DCNN [27] regards graph convolution as a diffusion process with graph convolution realized via the probability transition matrix between nodes.

GCN uses information regarding graph edges to aggregate node information for generating a new node representation, so the adjacency matrix representing the edge relationship is particularly important. In a traffic network graph, an adjacency matrix is usually constructed based on the distance or connectivity between nodes. DCRNN [8] models the traffic flow as a diffusion process on a directional graph, which constructs an adjacency matrix based on the distance map and uses diffusion graph convolution to extract the spatial representation. STGCN [9] builds an adjacency matrix based on the distance graph, with spectral domain GCN used to extract the spatial representation. HetGAT [16] builds an adjacency matrix based on the distance graph between nodes and the relationship between nodes and sites was also built and used GAT to extract the spatial representation. AST-MTL [18] builds an adjacency matrix based on the connectivity between nodes and the spatial representation was extracted using spectral domain GCN. A simple distance graph extracts the spatial relationship between nodes and it is easy to ignore the correlation between nodes. STSGCN [11] connects adjacent nodes to directly capture the local correlation between close nodes and their spatio-temporal neighbors, which placed limitations on capturing correlations between far nodes. HGCN [28] builds an adjacency matrix based on the distance graph and uses the distance between nodes to generate a regional graph for obtaining node similarity. DCNN was then used to extract the spatial representation, but the regional similarity tended to ignore single node differences. STGODE [17] and STFGNN [29] used DTW (Dynamic Time Warping) to compare similarities between time

series, but it is computationally expensive. AST-GCN [1], MS-net [13], and DMVST-Net [26] comprehensively modeled actual traffic data by integrating various external information such as POI (Point of Interest) distribution, weather, and holiday notifications, but these are difficult to obtain in many scenarios.

Traffic prediction models can be divided into two categories according to the acquisition of temporal dependencies, which are either RNNs-based [1,2,8,18] or CNNs-based [10,16,17,29]. Models based on RNNs extract temporal dependencies through RNN, Long Short-Term Memory (LSTM), and Gated Recurrent Unit (GRU). AST-GCN [1] and DCRNN [8] use the graph convolution operation to replace the original linear transformation in the recursive unit to obtain spatio-temporal information. GST-GAT [2] and TGC-LSTM [30] use LSTM [31] to extract temporal information. AST-MTL [18] uses GRU to obtain spatio-temporal information after graph convolution. However, when a traffic prediction data set is relatively large, the complex gating in RNNs will generate a large computation workload. As RNNs rely on the memory of the previous step, this makes it difficult to capture large traffic flow fluctuations during peak traffic conditions [9]. Some studies tend to use CNNs for temporal relationships in traffic prediction. CNNs-based studies use convolution operations across the temporal information to model temporal dependencies and expand the receptive field through dilated convolution to obtain a wider range of temporal relationships [20]. GWnet [10] and HGCN [28] use stacked 1D and 2D dilated convolutions after graph convolution to increase the temporal receptive field of the model and show better short-term prediction results. STGODE [17] uses 1D dilated convolution after graph convolution to extract temporal information. STFGNN [29] uses graph convolution and 1D dilated convolution with a large dilation rate to extract spatio-temporal information in parallel. HetGAT [16] uses graph attention convolutional after 1D and 2D dilated convolutions to extract temporal information.

Spatial and temporal modalities in traffic prediction usually use different methods to extract information, so the mutual fusion of spatio-temporal information plays a crucial role. STGCN [9] and HetGAT [16] use separate modules to sequentially capture spatial and temporal relationships, thereby splitting the association between spatial and temporal dimensions. STSGCN [11] and MS-Net [13] concatenate the separately obtained spatio-temporal relationships, and GWnet [10], STGODE [17], and AST-MTL [18] add the separately obtained spatio-temporal relationships, STFGNN [29] multiplies the separately obtained spatio-temporal relationships, but these simple methods are not conducive to establishing long-term spatio-temporal fusion.

### 3. Methodology

#### 3.1. Problem Definition

Traffic flow prediction is a time series problem in the traffic road network structure. Usually, the traffic flow of a certain time series in the future is predicted based on traffic flow from a certain historical time series. Our work mainly predicts the traffic speed data of the expressway network. We first define several basic concepts to express this prediction problem.

**Definition 1.** *Road network structure graph G: We use an undirected graph $G(V, E)$ to describe the topology of the road network, where $V = \{v_1, v_2, \ldots \ldots v_n\}$ is the set of all nodes in the graph, which are the sensors in the road network, and the i-th node in V is represented by $v_i$. $E \in V \times V$ is the set of all edges in the graph, which is the connection relationship between each sensor.*

**Definition 2.** *Feature matrix X: For any node $v_i$ in $G(V, E)$ we use $X_{v_i}$ to represent its feature value. The feature value of all nodes in $V \in \mathbb{R}^{N \times N}$ can be represented by matrix X which is the feature matrix. Then, the feature value of the node at a particular moment t can be expressed as $X_{v_i}^t$. If the current time is t, the traffic flow of the node $v_i$ in the graph G at time $t + 1$ can be expressed as:*

$$X_{v_i}^{t+1} = f\left(G; X_{v_i}^1, X_{v_i}^2, \ldots \ldots, X_{v_i}^t\right) \qquad (1)$$

where $f$ is the mapping function, $X\epsilon\mathbb{R}^{N\times C}$ is the traffic flow information of each sensor, and $N$ is the number of nodes, $C$ is the number of channels.

**Definition 3.** *Study question: We mainly study a mapping function $f$ which can map the traffic flow in the historical time series of all sensors to the traffic flow in a future time series. That means that for any node $v_i$ in $G(V,E)$, the traffic flow information $\hat{Y} = f(X) = (\hat{Y}_{v_i}^{t_{m+1}}, \hat{Y}_{v_i}^{t_{m+2}}, \ldots\ldots, \hat{Y}_{v_i}^{t_{m+n}})$ at a particular moment t for n time steps in the future can be predicted given historical traffic flow information $X = (X_{v_i}^{t_1}, X_{v_i}^{t_2}, \ldots\ldots, X_{v_i}^{t_m})$ with m historical time steps, as shown in* Figure 2.

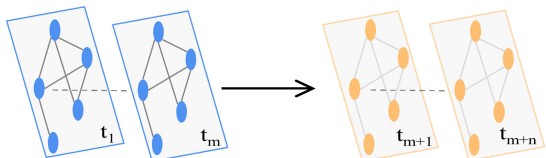

**Figure 2.** According to historical traffic flow time series graphs, future traffic flow time series graphs can be predicted (here *t* is time, *m* is the number of historical time steps, and *n* is the number of prediction time steps).

*3.2. Framework Overview*

The architecture of our model MFDGCN is shown in Figure 3. Transformation using the encoder–decoder structure has achieved great success in the field of natural language processing. As such, many studies have proposed that the deep learning architecture of the encoder–decoder is better at solving the sequence-to-sequence problem [32,33]. To obtain a better understanding of its short-term and long-term traffic prediction effects, we adopt the encoder–decoder structure in the network and stack several FDGCN layers in the encoder and decoder, respectively. Residual connections [34] and normalization are added to each layer to ensure that there is a training effect when the model network is deepened. To capture deeper spatio-temporal information, we use a multi-head attention mechanism between the encoder and the decoder and finally output the prediction result through the fully connected layer.

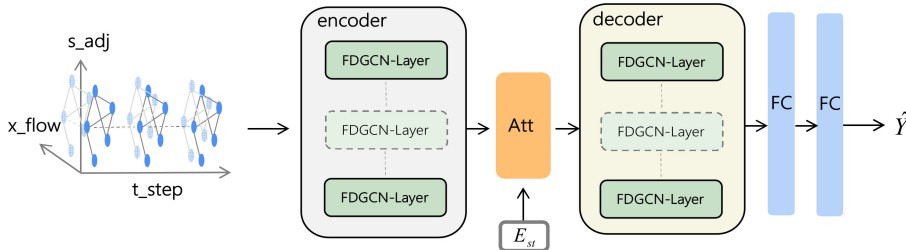

**Figure 3.** The framework of MFDGCN.

3.2.1. Association Graph Generation

Simple distance-based graphs can easily ignore related information between non-adjacent spatio-temporal information, so we propose a new spatio-temporal multi-association graph generation method. In this section, we will show how to use multiple graphs to capture static and dynamic spatio-temporal information in traffic.

Spatial static graph $G_{ss}(V, A_{ss})$: It is generated based on the adjacency of the spatial distance of nodes in the road network, where the spatial structure of the road network can be captured through $G_{ss}$. Two adjacent nodes in the spatial dimension are more likely to have similar flows because of the connectivity of the road network. We generate a matrix

$A_{ss}$ based on the node distance and set a certain threshold $\theta$ to filter further information. If there are nodes $v_i$ and $v_j$, $A_{v_i v_j}$ is the value of two nodes in matrix $A_{ss}$, which is defined as:

$$A_{v_i v_j} = \left\{ G, dis(v_i, v_j) \leq \theta \right\} \qquad (2)$$

where $dis(v_i, v_j)$ is the distance between nodes.

Spatial dynamic graph $G_{sd}(V, A_{sd})$: It searches for nodes with similar traffic flow to generate a graph according to the dynamic changes within a certain period of time in the road network. Spatial nodes with similar functions can be captured through $G_{sd}$. We select the $k$ nearest neighbor nodes to generate a matrix $A_{sd}$ according to the similarity of traffic flow of each spatial node in a certain period using KNN [22]. If there are nodes $v_p$ and $v_q$, $sim(v_p, v_q)$ is the similarity between the two nodes, and $A_{v_p v_q}$ is the value of the two nodes in the matrix $A_{sd}$ then:

$$A_{v_p v_q} = \begin{cases} 1, & sim(v_p, v_q) \epsilon k \\ 1, & v_p = v_q \\ 0, & otherwise \end{cases} \qquad (3)$$

The temporal dynamic graph $G_{td}(T, A_{td})$ ($T$ is time series) is based on dynamic changes in time to find time nodes with similar traffic flow to generate graphs. Time nodes with similar functions can be captured through $G_{td}$. For example, if there are two time points $T_p$ and $T_q$ which are both in the peak period, traffic flow may be equally large and the similarity is also large. We generate a matrix $A_{td}$ based on the similarity of traffic flow between time nodes and set a certain threshold $\theta$ to filter information that is far apart. Temporal static coding divides the time series into daily and weekly periods, which can capture the static adjacent time relationship of traffic flow.

### 3.2.2. Multi-Stage Hybrid Spatio-Temporal Fusion

Considering the complex and changing spatio-temporal information of the traffic network, we propose a multi-stage hybrid spatio-temporal fusion method. This effectively fuses temporal and spatial information at different levels and can capture longer-term spatio-temporal relationships.

In the early stage, we fuse the static and dynamic spatio-temporal graphs obtained from the association graph, respectively, and retain the original characteristics of the data. Relevant parameters are reduced as much as possible to obtain the spatial graph and temporal information. To obtain a better representation of the spatio-temporal feature, we use the Node2vec [35] method to perform node embedding operation on the spatial graph to obtain the spatial representation $E_s$. As for the temporal information, in order to avoid matrix sparsity, we use the One-Hot encoding operation to obtain the time representation $E_t$. The spatial and temporal representations are then fused to obtain the final spatio-temporal representation $E_{st}$, as shown in Figure 4.

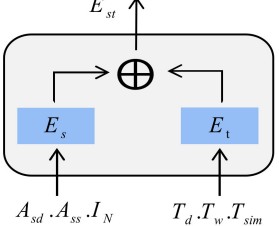

**Figure 4.** Static and dynamic spatio-temporal representation fusion (here $A_{sd}$ is the static distance graph of nodes, $A_{ss}$ is the dynamic similarity graph of nodes, and $I_N$ is the identity matrix; $T_d$ is the daily One-Hot static encoding of the time series, $T_w$ is the weekly One-Hot static encoding of the time series, and $T_{sim}$ is the dynamic similarity graph representation of the time series).

In the mid-stage, inspired by the structure of LSTM, the gate fusion mechanism is used to adjust the state of information flow in neural networks. We design an adaptive

gated spatio-temporal fusion mechanism that weights the different feature vectors of the multimodal input by dynamically generating adaptive weights. It can adaptively learn and control important features and assign larger weights to them, which can increase the feature weights required by the target and reduce the irrelevant feature weights. The adaptive gating structure is shown in Figure 5.

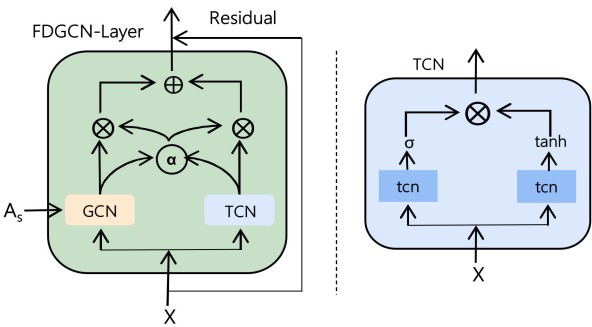

**Figure 5.** Adaptive gated spatio-temporal fusion mechanism.

When feature $X$ is input into the network, the adaptive gating adjustment parameter $\alpha$ receives feature vectors from different modes and weights the importance of hidden layer information of each time step to the target information. It then adaptively adjusts the output information required by the target. For the hidden network layer $l$, the calculation and weighting of the gate $\alpha$ is defined as:

$$H_s^{(l)} = G_\otimes(W_s \cdot X_s) \tag{4}$$

$$H_t^{(l)} = T_\otimes(W_t \cdot X_t) \tag{5}$$

$$\alpha = \sigma(\otimes(H_s^{(l)} + H_t^{(l)})) \tag{6}$$

$$H_{st}^{(l+1)} = H_s^{(l)} \cdot \alpha + H_t^{(l)} \cdot (1 - \alpha) \tag{7}$$

where $W \in \mathbb{R}^{C^{(in)} \times C^{(out)}}$ is the weight matrix, $X_s \in \mathbb{R}^{N \times C}$ and $X_t \in \mathbb{R}^{N \times C}$ are the input features of spatial and temporal information, respectively, $H_s^{(l)}$ is the output of the hidden layer after the graph convolution of $X_s$, $H_t^{(l)}$ is the output of the hidden layer after the temporal convolution of $X_t$. $\otimes$ is the convolution operation, $G_\otimes$ is the graph convolution operation, $T_\otimes$ is the temporal convolution operation, and $\sigma$ is the *sigmoid* activation function, and $H_{st}^{(l+1)}$ is the output result of the hidden layer $l$.

In the late decision stage, we use a multi-head attention mechanism to deeply fuse the results obtained in the early and middle stages again. For each head attention, we determine the importance of the spatio-temporal representation as predicted from the historical spatio-temporal representation and sum it with the weighted feature matrix after transformation. The multi-head attention results are then finally concatenated. For the hidden network layer $l$, the calculation of spatial and temporal attention fusion can be defined as:

$$a_{ij} = softmax((Q_j \cdot K_i^T) \cdot h^{-0.5}) \tag{8}$$

$$H_h^{(l)} = \sum_{x=1}^{n} a_{ij} \cdot V_w \tag{9}$$

$$Attention = concat(H_1^{(l)}, H_2^{(l)}, \ldots\ldots, H_h^{(l)}) \tag{10}$$

where $K_i$ represents the $i$-th vector in the historical spatio-temporal representation, $Q_j$ represents the $j$-th vector in the spatio-temporal representation to be predicted, and $V_w$ represents the $w$-th vector in the feature representation. $h$ is a constant and in our work we set $h$ to be the number of attention heads. $H_h^{(l)}$ is the attention output result of the $h$-head

hidden layer. $a_{ij}$ is the output of a *softmax* model, and the sum of the probability values is 1, which represents the importance of the $i$-th vector in the historical spatio-temporal representation to the $j$-th vector in the prediction spatio-temporal representation.

### 3.2.3. FDGCN Layer

In the MFDGCN, each FDGCN layer consists of graph convolutional network, temporal convolutional network, adaptive gating and residual connections [34]. The graph convolutional network obtains the spatial dependence between nodes by continuously aggregating neighborhood information, as shown in Figure 6.

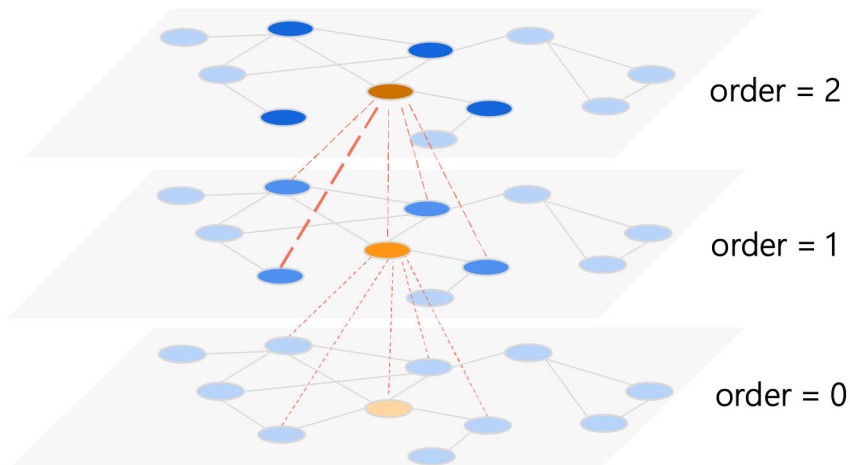

**Figure 6.** A GCN aggregates neighborhood information to obtain the spatial dependence between nodes (order = 1, the node aggregates its first-order neighbors; order = 2, the node aggregates its second-order neighbors).

Extraction of spatial dependencies: we refer to DCNN so that we can use spatially diffused graph convolution to extract spatial dependencies between nodes in the traffic network. DCNN regard graph convolution as a diffusion process, which assumes that information is transferred from one node to another adjacent node with a certain transition probability and that it is continuously diffused. If the traffic road network is regarded as graph $G(V, E)$, where $V \in \mathbb{R}^{N \times N}$ is the set of all nodes, $E$ is the set of edges, and $X \in \mathbb{R}^{N \times C}$ is the feature matrix. If $v_i, v_j \in V$, for the network hidden layer $l$, the diffusion graph convolution of node $v_i \rightarrow v_j$ is defined as:

$$z = f\left(W \odot P_{v_i \rightarrow v_j} X\right) \tag{11}$$

where $f$ is the mapping function, $W \in \mathbb{R}^{C^{(in)} \times C^{(out)}}$ denotes the weight matrix, $z \in \mathbb{R}^{N \times C}$ denotes the output. $P_{v_i \rightarrow v_j} \in \mathbb{R}^{N \times N}$ denotes the probability transition matrix, $N$ is the number of nodes, the $\odot$ operator represents element-wise multiplication, and finally output through the fully connected layer.

In our work, we use the adjacency matrix $A_s$ obtained from the spatial fusion graph with distance and spatial similarity as the probability transition matrix $P$ for diffusion convolution. Implicit representation of the previous layer is then calculated as a function of this layer. If given a node $v_i$ and its $p$-order neighbor set $v_p^*$ in the graph, for the network hidden layer $l$, we define the diffusion graph convolution of node $v_i \rightarrow v_p^*$ as:

$$H_{v_i \rightarrow v_p^*}^{(l)} = f(A_s H_{v_i \rightarrow v_p^*}^{(l-1)} W^{(l)}) \tag{12}$$

$$A_s = A_{sd} + A_{ss} + I_N \tag{13}$$

where $A_s \in \mathbb{R}^{N \times N}$, $A_{sd}$ is the static distance graph of nodes, $A_{ss}$ is the dynamic similarity graph of nodes, and $I_N$ is the identity matrix.

In our work, the diffusion order of the graph convolution is 2, and the node is diffused to its 2-order neighbors, then the diffusion convolution of the node $v_i$ is defined as:

$$H_{v_i}^{(0)} = X_{v_i} \tag{14}$$

$$H_{v_i \to v_1^*}^{(1)} = f(A_s H_{v_i}^{(0)} W^{(1)}) \tag{15}$$

$$H_{v_i \to v_2^*}^{(2)} = f(A_s H_{v_i \to v_1^*}^{(1)} W^{(2)}) \tag{16}$$

$$H_{v_i} = \sigma(H_{v_i}^{(0)} + H_{v_i \to v_1^*}^{(1)} + H_{v_i \to v_2^*}^{(2)}) \tag{17}$$

where $v_1^*$ and $v_2^*$ are the 1-order neighbor set and 2-order neighbor set of the node $v_i$, $\sigma$ is the *Relu* activation function.

Extraction of spatial dependencies. Temporal Convolutional Network (TCN) [36] is a model proposed in recent years with time series data processing ability. Its modeling ability on time series data sets is better than the recurring structure in a recurrent neural network, it is simple and effective without skipping cross-layer connections and can be computed in parallel. For the value of a hidden layer $l$ in the network at time $t$, it only depends on the value of the previous layer $l - 1$ at time $t$ and before, and cannot see the future. Therefore, it is more suitable for traffic prediction with time series characteristics. To keep each hidden layer the same length as the input layer, the temporal convolutional network increases channels via zero padding. To alleviate the complexity of the model caused by deepening the network, TCN introduces dilated convolutions to expand the receptive field of the convolution calculation.

In our work, we use temporal convolutional network to deal with temporal dependencies. For the input feature $X \in \mathbb{R}^{N \times C}$ and a filter $f : \{0, \ldots, k-1\}$, the temporal convolution operation $F$ of any element $s \in X$ of the network is defined as:

$$F(s) = \sum_{i=0}^{k-1} f \cdot X_{s-d \cdot i} \tag{18}$$

where $d$ is the dilation factor, $k$ is the size of the convolution kernel. Here, we set $k = 2$, and the above formula can be simplified as:

$$F(s) = f \cdot X_s + f \cdot X_{s-d} \tag{19}$$

The temporal convolution operation of any element $s$ is equal to the convolution of the element $s$ of the previous layer and the $(s - d)$-th element. In our work, the value of $d$ is different in each layer.

To enhance the extraction of complex temporal dependencies in the model, we add a gating mechanism to the temporal convolutional network. The *sigmoid* function is used to highlight strong relationships and filter weak relationships, with the *tanh* function used to control the data output between $(-1, 1)$. Amplification of important information is undertaken by multiplying the output results of two different activation functions. The final temporal convolutional network output is:

$$H = \tanh(F(a)) * \sigma(F(b)) \tag{20}$$

where $F(a)$ is the 2D TCN operation in the spatial and temporal dimensions, $F(b)$ is the 1D TCN operation in the temporal dimension, *tanh* and $\sigma$ are the activation functions.

## 4. Experiments and Discussion

In this section, we evaluate, compare, and analyze the experimental results of the proposed MFDGCN model, which compares eight basic and advanced traffic prediction baseline models when used with a real data set.

### 4.1. Data Set

We comprehensively evaluated the MFDGCN model on the real road network data set PeMS_BAY [8], which is shown in Figure 7. This data set contains the data of 325 sensors in the highway network collected by CalTrans Performance Measurement System (PeMS). We selected data at 30 s intervals for 6 months from 1 January 2017 to 31 May 2017. Before the experiment, we aggregated 30 s of data into a time step of 5 min as the traffic flow input feature $X$ of the model and generated an adjacency matrix $A_s$ based on the distance and traffic similarity of each sensor. For the traffic flow data, we used 70% of the data for training, 20% for testing, and 10% for validation. In our work, the number of time steps in an hour is set to 12, and on this basis, we sample features x and labels y by a sliding window with a step size of 1.

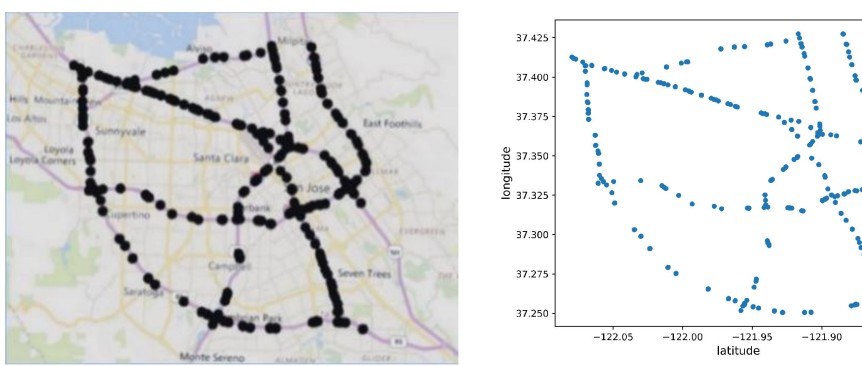

**Figure 7.** PeMS_BAY data set for traffic prediction. Black points mean location of sensors, each sensor measures traffic data every 30 s. PeMS_BAY constructed a road network based on the measured data.

### 4.2. Experimental Settings

We used PyTorch 1.10 to conduct experiments on GeForce RTX 2080Ti. The batch size was 32, the initial learning rate was 0.001, the decay was performed every 5 rounds, and the early stop mechanism was set during training. The number of FDGCN-layers was 8, and residuals were added to each layer. In the temporal convolutional network, the kernel size $k = 2$, and the dilation factor was $d = 1, 2, 1, 4, 1, 4, 1, 2$. In the graph convolutional network, the convolutional neighborhood order = 2. The number of attention heads was $h = 8$, and the time step $T$ was 12. We used the Adam optimizer [36] to train the model. The traffic prediction in our work is a regression problem, so in the performance evaluation stage, we adopted the regression commonly used evaluation metrics MAE, RMSE, and MAPE.

### 4.3. Baselines

In the experiments, we compared MFDGCN against the following eight baseline methods. HA [19]: The average value of all historical records in a certain period as the predicted value. VAR [20]: Used in the analysis of multivariate time series models, it can better reflect the fluctuation of traffic flow and reduce uncertainty. SVR [21]: Curve fitting using SVM and regression task analysis. DCRNN [8]: Traffic prediction using the encoder–decode framework and diffusion graph convolution. STGCN [9]: Designing neural networks with convolutional layers for spatio-temporal prediction with fewer parameters and faster training. STSGCN [11]: Design of multiple modules with different time periods to effectively capture heterogeneity in the local spatio-temporal graph. GWnet [10]: Stack spatio-temporal layers for better short-term predictions. GMAN [12]: Multi-attention mechanism modeling to solve the problem of poor long-term traffic prediction.

### 4.4. Experimental Results

This section shows the prediction results of MFDGCN on traffic flow and compares it with other baseline models. The results are shown in Table 1. We aggregated the MAE, RMSE, and MAPE metrics for the prediction results of the model over the next 15, 30, and 60 min on the PeMS_BAY data set, respectively. In this experiment, five-fold cross-validation was performed on the neural network model and the average value was taken as the final result.

**Table 1.** Comparison of 15, 30, and 60 min traffic prediction performance between MFDGCN and baseline models on the PEMS_BAY data set.

| Method | 15 min | | | 30 min | | | 60 min | | |
|---|---|---|---|---|---|---|---|---|---|
| | MAE | RMSE | MAPE | MAE | RMSE | MAPE | MAE | RMSE | MAPE |
| HA [19] | 2.88 | 5.59 | 6.80% | 2.88 | 5.59 | 6.80% | 2.88 | 5.59 | 6.80% |
| VAR [20] | 1.74 | 3.16 | 3.60% | 2.32 | 4.25 | 5.00% | 2.93 | 5.44 | 6.50% |
| SVR [21] | 1.85 | 3.59 | 3.80% | 2.48 | 5.18 | 5.50% | 3.28 | 7.08 | 8.00% |
| DCRNN [8] | 1.38 | 2.95 | 2.90% | 1.74 | 3.97 | 3.90% | 2.07 | 4.74 | 4.90% |
| STGCN [9] | 1.36 | 2.96 | 2.90% | 1.81 | 4.27 | 4.17% | 2.49 | 5.69 | 5.79% |
| STSGCN [11] | 1.44 | 3.01 | 3.04% | 1.83 | 4.18 | 4.17% | 2.26 | 5.21 | 5.40% |
| GWnet [10] | 1.30 | 2.74 | 2.73% | 1.63 | 3.70 | 3.67% | 1.95 | 4.52 | 4.63% |
| GMAN [12] | 1.34 | 2.82 | 2.81% | 1.62 | 3.72 | 3.63% | 1.86 | 4.32 | 4.31% |
| MFDGCN | 1.30 | 2.75 | 2.75% | 1.61 | 3.66 | 3.64% | 1.88 | 4.33 | 4.45% |

Table 1 shows that the non-neural network models (HA, VAR, SVR) perform poorly for traffic flow prediction, mainly because they cannot directly model spatio-temporal dependencies in the traffic flow. The neural network model has strong feature learning ability and can achieve good results in traffic flow prediction. Our MFDGCN model outperformed other baseline models in MAE, RMSE, and MAPE by comprehensively comparing long-term and short-term prediction effects.

As shown in the table, MFDGCN has a MAE that is 4% lower for a short-term prediction of 15 min and a RMSE that is 7% lower when comparing with the GMAN model with better long-term prediction performance. MFDGCN has a MAE that is 7% lower for a long-term prediction of 60 min and a RMSE that is 19% lower when comparing with the GWnet model with better short-term prediction performance. In the comprehensive comparison, the MFDGCN model goes further when compared to the GWnet and GMAN models. It fuses spatio-temporal features better, obtains richer spatio-temporal dependencies, and achieves better results in both short-term and long-term traffic prediction. As the prediction duration increases, the long-term prediction performance of each model will decline, but the decline for MFDGCN is smaller and its advantage is more obvious.

Compared with STGCN, DCRNN, GWnet, and GMAN modeled using a simple distance graph and STSGCN modeled with a local node correlation graph, MFDGCN uses dynamic and static fusion graph modeling in spatial and temporal dimensions. This can not only capture relevant information from local nodes in the road traffic network, but also capture relevant information from global nodes. It can learn the similar relationship between non-adjacent nodes and improve prediction performance. Compared with STGCN that splits the spatio-temporal association, STSGCN concatenates spatio-temporal information and GWnet adds the stacked spatio-temporal information, MFDGCN uses an adaptive gating mechanism to fuse the convolutional spatio-temporal information. It then fuses the embedded spatio-temporal information through a multi-head attention mechanism, which can capture longer-term spatio-temporal relationships and achieve better long-term prediction performance.

MFDGCN shows better short-term prediction performance by using multi-layer stacked and dilated temporal convolutions. It is 4% lower in MAE for a short-term prediction of 15 min and 1% lower in MAE at 30 min when compared with GMAN where a transformer is used. However, MFDGCN is slightly lacking in long-term prediction when

compared with GMAN and the MAE of GMAN at 60 min for a long-term prediction is 2% higher. In general, the comprehensive performance of MFDGCN is better.

The comparison of the prediction performance of each model at each time step on the PeMS_BAY data set is shown in Figure 8. Figure 9 shows the comparison of the truth and predicted values of MFDGCN for a certain day on the PeMS_BAY data set.

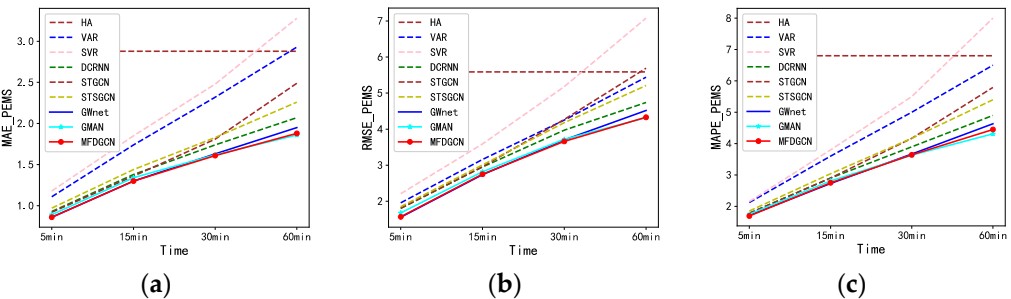

| (a) | (b) | (c) |

**Figure 8.** Comparison of prediction performance at each time step on the PeMS_BAY data set. (**a**) MAE values comparison; (**b**) RMSE values comparison; (**c**) MAPE values comparison.

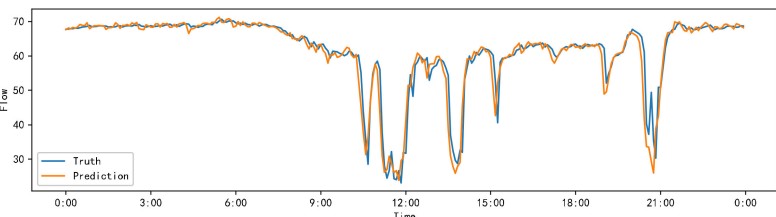

**Figure 9.** Comparison of the truth value and the predicted value.

### 4.5. Effects of Components and Parameters

We compared the proposed multi-association graph generation and multi-stage fusion methods, verified the effectiveness of each component in MFDGCN, and compared the selection of important model parameters. We removed the spatial similarity graph in the multi-association graph and named the model No-simg. We also removed the gate in the multi-stage fusion and named the model No-gate. Comparing them with the MFDGCN model, the MAE values of the prediction results at 15 min are shown in Figure 10. The results show that stronger features can be obtained by using the multi-association graph and the spatio-temporal multimodal fusion module shows better performance.

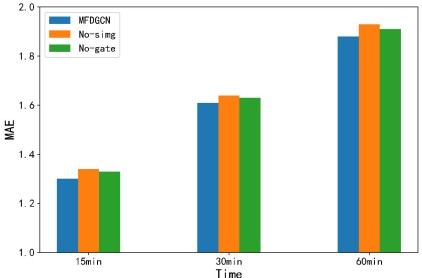

**Figure 10.** Comparison of MAE values of prediction results of MGSTCN, No-simg, and No-gate models at 15, 30, and 60 min on the PeMS_BAY data set.

We used *L* to represent the number of layers of FDGCN, fixed each component in FDGCN, and tested the model prediction results when *L* was set to 2, 4, 6, 8, and 10. The results are the average of all MAE values within an hour. As shown in Figure 11a, the model works best when *L* = 8. The results show that the model needs a certain depth to obtain richer spatio-temporal information, but too much depth will influence the model effect, and parameter selection needs to be made according to different data and models.

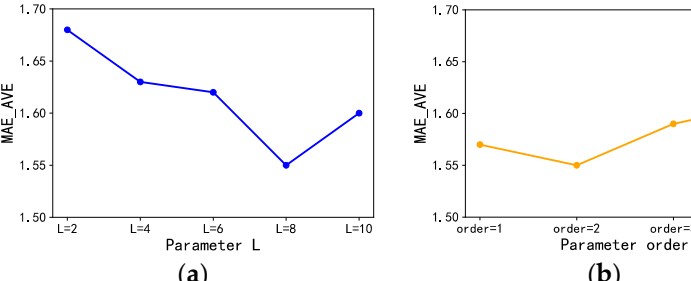

**Figure 11.** The average value of MAE when the parameter takes different values. (**a**) Parameter *L*; (**b**) parameter order.

We fixed the number of FDGCN layers at *L* = 8 and tested the model prediction results when the graph convolutional network neighborhood order was set to 1, 2, 3, and 4. The results are the average of all MAE values within an hour. As shown in Figure 11b, the model works best when order = 2. The results show that graph convolution is not as large as possible when selecting neighborhoods and appropriate parameters should be selected according to the different data and models.

## 5. Conclusions

In this paper, we proposed a new deep learning model Multi-Stage Spatio-Temporal Fusion Diffusion Graph Convolutional Network (MFDGCN) based on the diffuse convolution method to solve the problem with deep spatio-temporal dependencies being easily ignored in road network traffic prediction based on simple distance maps and spatio-temporal fusion methods. MFDGCN combines the graph convolutional network for capturing spatial dependencies and the temporal convolutional network for capturing temporal dependencies. It generates multi-association graphs to learn similar relationships between non-adjacent nodes and uses a multi-stage hybrid spatio-temporal fusion mechanism to capture longer-term spatio-temporal associations and mine deep spatio-temporal dependencies. From evaluation with real-world traffic data, MFDGCN shows good performance in both long-term and short-term prediction of traffic flow. It can also be used to solve other similar spatio-temporal prediction problems. In future work, we will continue to optimize the network structure as well as parameters and further study the dynamic graph structure problem to make the neural network more flexible.

**Author Contributions:** Conceptualization, Z.C. and H.J.P.; methodology, Z.C. and H.J.P.; software, Z.C. and J.Z.; analysis, G.N. and H.J.P.; resources, Z.C. and H.J.P.; data curation, G.N. and H.J.P. visualization, Z.C. and J.Z.; supervision, G.N. and H.J.P. All authors have read and agreed to the published version of the manuscript.

**Funding:** Not applicable.

**Institutional Review Board Statement:** Not applicable.

**Informed Consent Statement:** Informed consent was obtained from all subjects involved in the study.

**Data Availability Statement:** The data presented in this study is a public data set that can be downloaded from the public data provider https://pems.dot.ca.gov.

**Conflicts of Interest:** The authors declare no conflict of interest regarding the publication of this paper.

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
