# Peer review of "MFDGCN: Multi-Stage Spatio-Temporal Fusion Diffusion Graph Convolutional Network for Traffic Prediction"

_applsci, doi:10.3390/app12052688_

Round 1
Reviewer 1 Report
A new deep learning model multi-stage spatio-temporal fusion diffusion graph convolutional network (MFDGCN) is presented to better obtain the comprehensive effect of long-short term prediction and capture the deep and complex spatio-temporal information. From my view, this paper is well organized and the proposed method is valuable for this research filed. After reviewed this paper, there are some questions as follows.
(1) In the abstract, the author should highlight the specific problems to be solved in this study at the beginning, and then lead to the solutions. At present, the description is not clear.
(2) The literature review is poor in this paper. You must review all significant similar works that have been done. Also, review some of the good recent works that have been done in this area and are more similar to your paper. For example, 10.1007/s12559-021-09871-4; 10.1109/TCYB.2020.3033005; 10.1016/j.ins.2021.11.052 and so on.
(3) What is meaning of t in Figure 2?
(4) Figure 1, Figure 7, …. are not clear, please revise them to be clear.
(5) The authors need to interpret the meanings of the variables.
(6) In the part of results and discussion, the author should compare the relevant work in this field with the results of this paper to confirm the effectiveness of this study.
(7) What are the advantages and disadvantages of this study compared to the existing studies in this area?
(8) There are some grammatical mistakes and typo errors.
Reviewer 2 Report
In the present manuscript, the authors propose a new deep learning model named Multi-Stage Spatio-temporal Fusion Diffusion Graph Convolutional Network (MFDGCN) which they use for traffic prediction. The model contains several complex features that allow better long and short-term traffic prediction than existing models. The authors further compare their model with commonly used Machine learning procedures, i.e. HA, VAR, SVR, DCRNN, STGCN, STSGCN, Graph WaveNet, and GMAN. I find this comparison very useful for the reader.
In summary, I do think it will be of interest to the community.
However, I have a specific comment regarding the presentation of the manuscript. I think that the authors should take an extra effort and improve the clarity of the manuscript and most important, the comparison with related works.
In the introduction, for example, the authors keep talking about “Many studies” used this or that approach but they fail to provide citations or discuss in more detail the connection with their work. Several examples can be found on page 2. I would recommend a revision here.
A minor comment is that authors should avoid using “etc.” and spell out all items when they list something (see examples on page 3).
POI (Point of interest) should be introduced before its first inclusion in the text (before page 2).
Reviewer 3 Report
Below are my comments and suggestions.
1. It seems that the authors missed some recent relevant references. References need to be updated.
2. Please add a section and discuss about the advantages and disadvantages of the existing approaches.
3. It is essential to make sure that the manuscript reads smoothly- this definitely helps the reader fully appreciate your research findings.
Round 2
Reviewer 1 Report
According to the revised paper, I have appreciated the deep revision of the contents and the present form of this manuscript. But there is still a little content, which need be revised according to the comment of reviewer in order to meet the requirements of publish. A number of concerns listed as follows:
(1) The authors need to interpret the meanings of the variables.
(2) More equations are necessary to explain the proposed method.
(3) Please highlight your contributions in introduction.
(4) Authors revised the literature review, but it still can not meet the requirements. Please further revise it according to the comments in the first review.
Round 3
Reviewer 1 Report
This is ok.